# Quantitative time-resolved analysis reveals intricate, differential regulation of standard- and immuno-proteasomes

Juliane Liepe[1], Hermann-Georg Holzhütter[2], Elena Bellavista[3], Peter M Kloetzel[2], Michael PH Stumpf[1]*, Michele Mishto[2,4]*

[1]Centre for Integrative Systems Biology and Bioinformatics, Department of Life Sciences, Imperial College London, London, United Kingdom; [2]Institut für Biochemie, Charité - Universitätsmedizin Berlin, Berlin, Germany; [3]Department of Experimental, Diagnostic and Specialty Medicine, Alma Mater Studiorum, University of Bologna, Bologna, Italy; [4]Luigi Galvani, Alma Mater Studiorum, University of Bologna, Bologna, Italy

**Abstract** Proteasomal protein degradation is a key determinant of protein half-life and hence of cellular processes ranging from basic metabolism to a host of immunological processes. Despite its importance the mechanisms regulating proteasome activity are only incompletely understood. Here we use an iterative and tightly integrated experimental and modelling approach to develop, explore and validate mechanistic models of proteasomal peptide-hydrolysis dynamics. The 20S proteasome is a dynamic enzyme and its activity varies over time because of interactions between substrates and products and the proteolytic and regulatory sites; the locations of these sites and the interactions between them are predicted by the model, and experimentally supported. The analysis suggests that the rate-limiting step of hydrolysis is the transport of the substrates into the proteasome. The transport efficiency varies between human standard- and immuno-proteasomes thereby impinging upon total degradation rate and substrate cleavage-site usage.

**\*For correspondence:**
m.stumpf@imperial.ac.uk (MPS);
michele.mishto@charite.de (MM)

**Competing interests:** The authors declare that no competing interests exist.

**Reviewing editor**: John Kuriyan, Howard Hughes Medical Institute, University of California, Berkeley, United States

## Introduction

Cells can be usefully thought of as communities of molecular machines that work in concert to maintain cellular function (*Nurse, 2003*). Metabolism, gene expression, protein production and degradation, energy production, DNA replication, cell division all have their own repertoire of associated proteins and macromolecular assemblies, and are controlled by complex and highly dynamic signalling and regulatory networks; this picture is complemented by a slew of structural proteins that provide cell walls, nuclear membranes, endoplasmic reticulum and other structures of the eukaryotic cell.

But while advances in imaging and microscopy technology have allowed us to glean some insights into the structures and even dynamics at the molecular level (*Liepe et al., 2012*), much of our knowledge about the function of for example, protein production at ribosomes, or degradation at proteasomes, is based on indirect observations, or on data collected at some experimentally convenient equilibrium state. Increasingly time-resolved data can also be collected, and this type of data typically provides more detailed insights into the molecular mechanisms at work. To interpret such data, mathematical modelling coupled to state-of-the-art statistical inference thus becomes a necessity (*May 2004*; *Buchholz et al., 2013*; *Gerlach et al., 2013*) to make sense of data and design better, more discriminatory experiments.

**eLife digest** Cells have to be able to reliably destroy or remove molecules from their interior that they no longer need. Structures called proteasomes play a central part in this complex process by cutting up and digesting proteins. Mammals have several different types of proteasomes, each made up of several protein 'subunits'. For example, when a cell experiences inflammation some proteasomes change some of their subunits and form an immuno-proteasome. These immuno-proteasomes tend to break down proteins more quickly than 'standard' proteasomes, but it was not clear how they are able to do so.

Liepe et al. have now combined experiments and mathematical modelling to construct a detailed model of proteasome activity. The model shows that protein transport into and out of the proteasome chamber are the steps that limit how quickly the proteasomes can break down proteins. Furthermore, these transport processes are also to a large extent responsible for the different rates at which standard and immuno-proteasomes process proteins. Liepe et al. were also able to confirm the existence of regulatory sites within the proteasome, and describe how these are arranged.

Problems that alter the rate at which proteasomes break down proteins have been linked to tumors and neurological and autoimmune diseases. Liepe et al.'s model opens up the ability to study how the proteasome's activity is affected by drugs and therefore makes it easier to investigate ways of interfering with this activity for therapeutic purposes.

Here we apply a modelling approach to a detailed mechanistic analysis of proteasomes, which are at the core of the ubiquitin-proteasome system and responsible for the destruction of the majority of the cytoplasmic proteins. Proteins are usually tagged by E1-E2-E3-E4 enzyme-mediated poly-ubiquitination, carried into the proteasome proteolytic chamber where they are fragmented and, ultimately, expulsed (*Schwartz and Ciechanover, 2009*). Proteins, and especially oxidised proteins—that is, those that reflect the presence of cellular stress conditions—can also be degraded by 20S proteasomes without prior poly-ubiquitination (*Aiken et al., 2011*; *Pickering and Davies, 2012*; *Ben-Nissan and Sharon, 2014*; *Höhn and Grune, 2014*). In addition to stress response, the 20S proteasomes present in cells (*Fabre et al., 2015*) have been shown to regulate the abundance of a host of signalling molecules involved in cell cycle progression, cellular growth control and oncogenesis (*Ben-Nissan and Sharon, 2014*). Additionally, 20S proteasomes form the core of the structure of 26S proteasomes and we cannot expect to understand the latter if we do not understand the mechanisms that determine the activity of the former.

Poly-ubiquitin tagging (when it occurs), transport and peptide-bond hydrolysis regulate protein half-life and thereby affect the majority of metabolic processes in the cell. Despite considerable biochemical and structural efforts, the mechanisms of proteasomal action, in particular the causes and extent of differences in substrate-specific proteolysis between different proteasome isoforms, are only incompletely understood. In this study we combine carefully designed experimental assays with detailed modelling to shed light on these mechanisms.

There is a dearth of mechanistic analyses of proteasome functions, and the complexity of polypeptide hydrolysis has precluded detailed analysis (*Liepe et al., 2014a*). The most direct insights come from carefully chosen short fluorogenic peptides, which have a single proteasome-catalyzed cleavage site, which in turn allows us to follow the kinetics in real time. They have also been used (exclusively) to measure the proteasomal activity in cellulo (e.g., *Kapeta et al., 2010*; *Chondrogianni et al., 2015*; *Peters et al., 2015*; *Pickering et al., 2015*), despite the fact that they lack the complexity of 'real' proteins and they do not recapitulate proteasome proteolytic activities towards polypeptide chains (*Mishto et al., 2014*). Nevertheless, they provide arguably the most promising class of leads of proteasomal inhibitors (*Bellavista et al., 2013*; *Kisselev and Groettrup, 2014*). Furthermore, kinetic and structural analyses rely on their structural and biophysical characteristics (*Gaczynska et al., 1993*; *Kisselev et al., 2002*, *2003*, *2006*; *Osmulski et al., 2009*); we, too, use them here for our analysis, which is based on a representative set of such peptides. As we show below the insights gained from the short fluorogenic peptides are borne out by further analysis of polypeptides.

The mechanistic analyses performed here also incorporate known structural features into a mathematical model of proteasome action. The 20S proteasome consists of four stacked seven membered rings (denoted by $\alpha_7\beta_7\beta_7\alpha_7$). These rings form three interconnected cavities, including a

pair of antechambers through which substrates are passed before reaching the central catalytic chamber. Antechamber walls are not merely structural, but interact actively with the substrates by altering their properties and keeping them accessible for hydrolysis (*Ruschak et al., 2010*). These proteasome chambers can also store two or more proteins in order to enable continuous degradation (*Hutschenreiter et al., 2004*; *Sharon et al., 2006*). The central chamber contains six subunits (two β1, β2 and β5 subunits) that catalyse the peptide-bond hydrolysis and peptide splicing after binding of the polypeptide substrates nearby their active N-terminal Thr (*Vigneron et al., 2004*; *Borissenko and Groll, 2007*; *Mishto et al., 2012*).

20S proteasomes modify their conformations and thus in turn their activity upon peptide-bond hydrolysis (*Osmulski et al., 2009*; *Ruschak and Kay, 2012*), binding of regulatory complexes such as 11S or 19S to the proteasome α rings (leading to the formation of PA28-capped proteasomes and 26S proteasomes, respectively) (*Dick et al., 1996*; *Emmerich et al., 2000*; *Köhler et al., 2001*; *Ruschak and Kay, 2012*; *Raule et al., 2014*), and activation of non-catalytic modifier sites (*Schmidtke et al., 2000*; *Kisselev et al., 2002*, *2003*) whose location in the proteasome remain unknown (*Liepe et al., 2014a*).

In mammals different 20S proteasome isoforms exist, which carry different catalytic β subunits. Upon an inflammatory stimulus such as IFN-γ the catalytic standard β1, β2 and β5 subunits peculiar of the standard proteasome (s-proteasome) are replaced by the immuno-subunits β1i, β2i and β5i in the newly synthesized immunoproteasome (i-proteasome). Proteasome isoforms degrade substrates with different rates, but generate the same peptide pool (*Mishto et al., 2014*). And while there exist differences between substrates, the 20S i-proteasome has been shown often to have a higher polypeptide degradation rate than its standard counterpart (*Bellavista et al., 2013*). Why s- or i-proteasomes should degrade specific substrates more rapidly is a matter of ongoing debate. Different steps of the proteolysis process may be responsible for isoform specific kinetics: transport of the substrate through the gate and the antechambers; binding to substrate binding sites and ensuing peptide-bond hydrolysis (*Huber et al., 2012*; *Arciniega et al., 2014*); and the release from the substrate binding sites and finally from the proteasome gate. The question is also of direct medical relevance: different degradation kinetics of specific substrates by s- and i-proteasomes, or i-proteasome carrying genetic polymorphisms/mutations, have been suggested to be involved in a variety of pathologies (*Basler et al., 2013*; *Bellavista et al., 2013*). Furthermore, i-proteasome deficient mice show altered proteasome-dependent kinetics of pathogen epitope generation (*Basler et al., 2013*). Such different kinetics could, for example, lead to the remodelling of immunodominance of viral epitopes (*Zanker et al., 2013*) or to the lack of CD8[+] T cell-mediated response towards specific viral antigens (*Deol et al., 2007*).

As we will show below, our framework allows us to start from simple models of proteasome function, and identify which mechanistic shortcomings they exhibit. We focus on the dynamics of the digestion of a set of exemplar peptides, and show that this process is regulated carefully and by three distinct mechanisms, that act in concert to regulate proteasomal activity. Here considering different mechanisms, coupled to state-of-the art statistical model selection techniques (*Kirk et al., 2013*), allows us to elucidate steps in the proteasomal dynamics that cannot be probed directly through experiments. Model selection naturally extends the conventional hypothesis testing approach, and by accounting for uncertainty fairly and correctly, can be applied to systems with many unknown kinetic parameters (which are, in any case, inferred together with their respective uncertainties) (*Liepe et al., 2013*; *Babtie et al., 2014*).

## Results

### Mechanistic modelling of proteasome proteolysis

#### Rate of product formation changes over time

The majority of mathematical models (*Schnell and Maini, 2003*) describing enzymatic activity are Michaelis-Menten-type (MM) models, that is, they describe the dependency of the initial reaction speed on the initial substrate concentration. Such models can capture allosteric effects and have for example, been applied to investigate the action and dynamics of specific inhibitors. However, they assume that the initial reaction velocity is constant.

The in vitro digestion of short fluorogenic peptides by purified mouse 20S proteasomes reveals, however, substrate inhibition at high substrate concentrations (*Figure 1A*), which is not compatible with the classical MM model. Furthermore, we find that proteasome proteolytic activity, that is, reaction velocity, changes over time as the product accumulates in non-linear fashion (*Figure 1B*). For instance, the reaction velocity for the substrate Suc-LLVY-MCA increases over time (*Figure 1B*); this temporal profile of proteasome activity is not due to its permanency at 37°C since Suc-LLVY-MCA degradation kinetics of the mouse proteasomes prior storing for 18 hr at 37°C or 4°C do not differ significantly (*Figure 1—figure supplement 1A*).

Similarly, we detect proteasome activity inhibition at high substrate concentrations (*Figure 1—figure supplement 1B*), and non-linear accumulation of product over time (*Figure 1—figure supplement 1C*), when using protein homogenate of human T2 cells, which contains 20S proteasome as well as proteasome regulatory complexes such as 19S and PA28.

In agreement with the data obtained by using the short fluorogenic substrates, we also observe that the frequency of peptide-bond hydrolysis after some cleavage sites of polypeptide substrates varies over time (*Figure 1C,D*). The changing substrate cleavage site usage over time might be the result of substrate depletion that leads to the re-entry and cutting of peptide products, thus acting as competitive inhibitors. This would lead to a decrease of the average length of the peptides products due to further cleavages and thus further shortening of the initial products. However, we observe only a slight reduction in the average length of the peptide

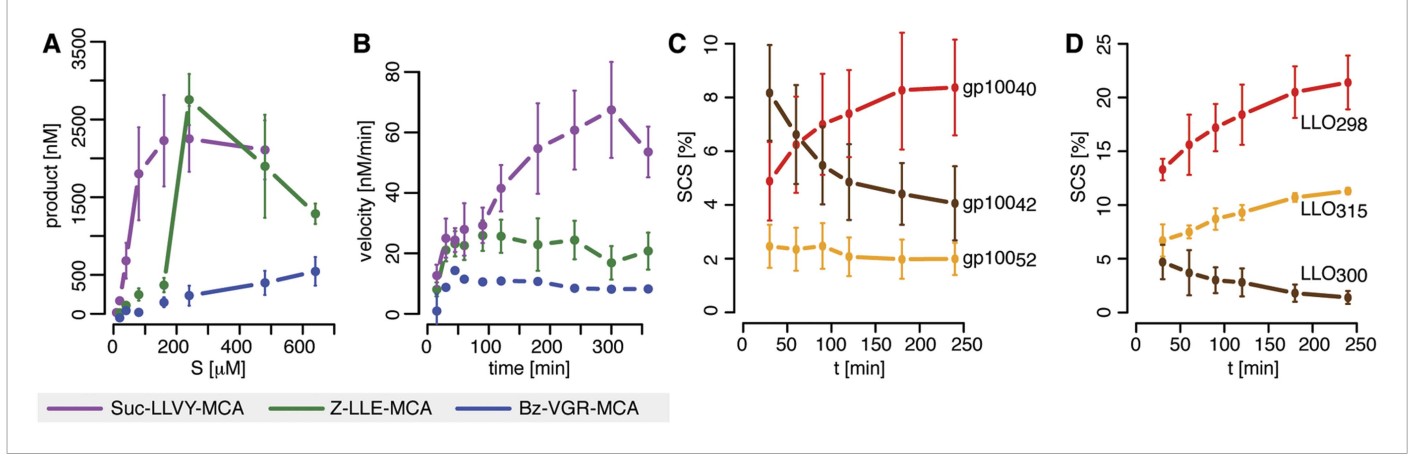

**Figure 1**. Velocity and specific cleavage site usage by mouse proteasomes varies over time. (**A**) The amount of products generated after 6 hr was measured for different initial substrate concentrations of the short fluorogenic substrates Suc-LLVY-MCA, Bz-VGR-MCA and Z-LLE-MCA by purified 20S mouse liver proteasome. (**B**) The reaction velocity [nM/min] of the same substrates (480 µM) as in (**A**) by purified mouse 20S proteasome was measured over time. (**C**, **D**) Cleavage rate (pmol peptide-bond hydrolysed/[min•mg proteasome]) after the residues gp100$_{40}$ (Arg), gp100$_{42}$ (Lys) and gp100$_{52}$ (Trp) of the synthetic polypeptide gp100$_{35-57}$ (**C**) as well as LLO$_{298}$ (Tyr), LLO$_{300}$ (Arg) and LLO$_{315}$ (Val) of the synthetic polypeptide LLO$_{291-317}$ (**D**) by mouse proteasome. Peptide product amount and site-specific cleavage strength (SCS) was computed by applying QME to each time point of the in vitro kinetics. Values are the mean and the SD of two independent experiments.

The following figure supplements are available for figure 1:

**Figure supplement 1**. Proteasome dynamics are not modified over time because of permanency at 37°C or product re-entry and further processing.

**Figure supplement 2**. Schematic of the substrate inhibition model.

**Figure supplement 3**. Michaelis–Menten and substrate-inhibition models do not describe the short fluorogenic peptide degradation by mouse proteasome.

**Figure supplement 4**. Gate opening by Rpt peptides override the enhancing effect mediated by LLVY peptide.

**Figure supplement 5**. Substrate inhibition effect is evident for Z-LLE-MCA degradation kinetics.

products in the polypeptide degradations, only at late time points, and only when less than 50% of the substrate is still intact (*Figure 1—figure supplement 1D,E*). By contrast, the changes in cleavage site usage are already becoming evident at early time points (*Figure 1C,D*). Therefore substrate depletion, product re-entry or further cleavage cannot explain variation in cleavage site usage over time.

To understand the proteasomal mechanisms in detail we therefore investigate product formation under carefully designed experimental conditions (*Liepe et al., 2013*), and measure the degradation of Suc-LLVY-MCA, Bz-VGR-MCA and Z-LLE-MCA over time by mouse proteasome for different initial substrate concentrations. Based on these data sets (available on Dryad; *Liepe et al., 2015*) we then develop a set of increasingly detailed mathematical models of proteasomal peptide hydrolysis (*Figure 2—source data 1*) and apply a Bayesian model selection (*Kirk et al., 2013*) framework (which also ensures a level of parsimony: i.e., models that are more complicated are only preferred if they are capable of capturing the data significantly better than simpler models) to elucidate proteasome action. As a by-product we also obtain parameter estimates (including an assessment of the corresponding uncertainty).

## Classical enzyme kinetic models fail to describe the time course of proteasomal peptide hydrolysis

A useful mathematical model needs to explain the time course of product formation over time. Here, we focus especially on the inhibition of product formation at high substrate concentrations (which can be best seen in dose-response curves) and on the increase of the reaction velocity at early time points (which can be seen on the individual time series).

Substrate degradation involves: binding of substrate close to the active site; peptide bond hydrolysis; and release of the products from the active site. This scheme is often assumed for enzymatic reactions and forms the basis of the MM model (*Figure 2A*); but it explains neither the substrate inhibition nor the increasing reaction velocity.

In the short fluorogenic peptide assay the product has the same amino acid sequence as the N-terminal part of the substrate (the C-terminal part contains the MCA-group that is cleaved off). Because of this the product itself can bind tightly to the active site without being further processed and thereby block the cleavage of further substrates (product inhibition). We employ the two-site-modifier scheme of *Schmidtke et al. (2000)*, who have already argued against the use of MM-types model for analysing proteasome function, and we adapt it to allow for product and substrate inhibition (*Figure 2B* and *Figure 1—figure supplement 2*). However, the substrate inhibition model still fails to reproduce our data, which can be clearly seen in the case of Z-LLE-MCA degradation (*Figure 1—figure supplement 3A,B*, blue curves). The intermediate complexes are assumed to be in quasi-steady state (which holds for all proteases studied so far), which allows us to reduce the overall complexity of the mathematical description without loss of information (*Sanft et al., 2011*; *Grima et al., 2014*).

Because of the observed increase in reaction velocity over time until 90–120 min, we next investigate if a positive feedback loop could cause this. Two distinct mechanisms are possible: either (i) the product enhances binding to the active site; or (ii) the product increases the peptide-bond hydrolysis rate (*Figure 1C,D*). Even though both mechanisms result in an increased reaction velocity over time, they cannot explain substrate inhibition and fail to reproduce our experimental data (*Figure 1—figure supplement 3C,D*). Because of this we require and develop a more complex model that accounts for the specific steps of proteasomal peptide hydrolysis.

## Substrate transport is a crucial step in modelling peptide hydrolysis

The models considered so far do not account for the proteasome structure where active sites are buried inside the proteasome's inner cavity; substrate must enter via the proteasome gate and move along the inner cavity until it reaches the active site. This structural organisation profoundly affects proteasome dynamics.

Previous in silico studies (*Liepe et al., 2014a*) had assumed that substrate molecules enter and leave the proteasome by diffusion. However, the proteasome gate and the interior surface of the proteasome chambers have strong partial charges, and substrate molecules have been shown to interact at least with the latter, and a transport model based on free diffusion cannot capture the observed data (*Figure 2E*).

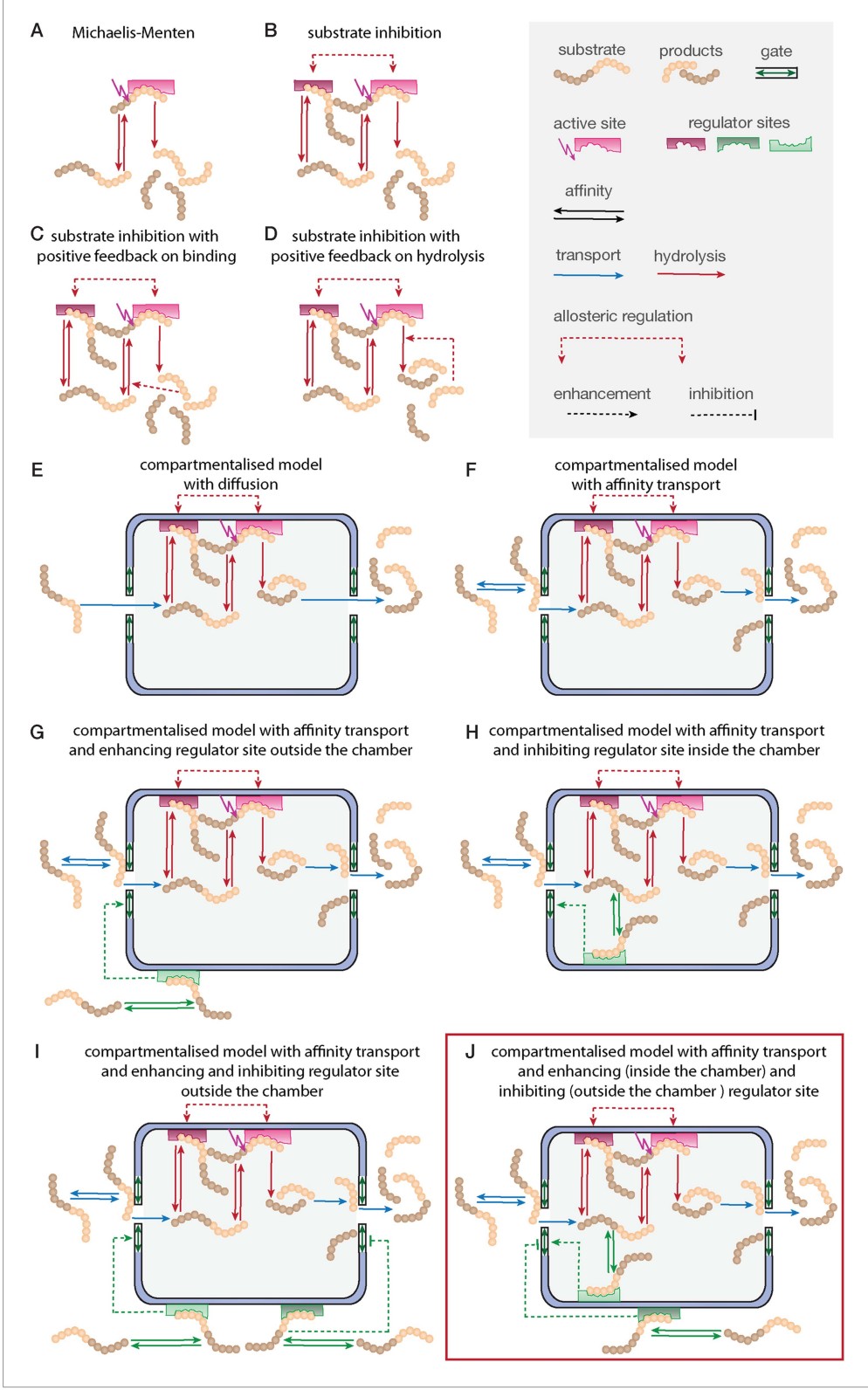

**Figure 2**. Development of the mathematical model. Schematics of the developed and tested models. Reactions involved in peptide-bond hydrolysis are indicated in red, steps involved in substrate and product transport are indicated in blue and the regulation of the transport is indicated in green. Models in (**A**–**D**) are without the proteasome as a separate compartment, while (**E**–**J**) are compartmentalised models. Note, for simplicity the

*Figure 2. continued on next page*

*Figure 2. Continued*

schematics contain only one active site (instead of the two copies for each active site). Furthermore peptides can enter and leave the proteasome chamber through both gates.

The following source data and figure supplements are available for figure 2:

**Source data 1**. Compartment models in SBML format.

**Source data 2**. List of mathematical model parameters.

**Source data 3**. List of mathematical model species.

**Figure supplement 1**. Compartmentalised models with affinity transport enhancing regulatory site(s) inside the chamber and substrate inhibitory site(s) best fit the experimental data.

**Figure supplement 2**. Compartmentalised model with affinity transport and enhancing regulatory site(s) inside the chamber and inhibiting site(s) outside the chamber fit the experimental data best.

Before substrate enters the 20S core, it needs to be located close to one of the two proteasome gates (*Figure 2F*). Substrate affinity to the gate is described by the parameters $k_{on}$ and $k_{off}$. After entry into the central chamber through the antechamber ($v_{in}$) the substrate can bind to an active site. The number of substrate molecules that can enter the chambers is determined by the maximum capacity ($C$—number of molecules that can be allocated in the proteasome chamber). This number depends on the volume of the proteasome chamber and the volume of the molecules. Since different substrate molecules differ in volume, the maximal capacity is expected to differ as well. The rate $v_{in}$ is dependent on how many molecules are already located inside the proteasome chamber. If the number of molecules inside the chamber reaches the maximal capacity $C$, the rate by which new molecules enter the chamber is decreased. Since the exact mechanism for this process is unknown we formulate a heuristic expression that decreases the rate as a function of the number of molecules inside the chamber and the maximal capacity (see *Supplementary file 1*, section 2).

To exit the proteasome products and unprocessed substrates need to move from the central chamber through the antechamber, and then to the gate ($\tau$), where they can exit the proteasome ($v_{out}$). Both unprocessed substrates and products can re-enter the proteasome. Substrates and products can enter and leave the proteasome through the same gate or through different gates. In the model we include both gates, which are described by an outer site (G1) and an inner site (G2). Note, in the schematics of *Figure 2* we show substrate entry always on the left hand side and substrate release always on the right hand side for simplicity. This does not indicate two distinct gates with distinct characteristics.

This proposed transport model is the simplest realistic representation of the biophysical transport mechanism; the possible bio-molecular interactions are summarised by rates ($k_{on}$, $k_{off}$, $v_{in}$, $\tau$, $v_{out}$), see *Figure 2—source data 2*.

## Peptide transport is regulated through open/closed gate conformation

The proteasomal hydrolysis rate changes over time and the kinetics vary between different fluorogenic peptides or synthetic polypeptides (*Figure 1B–D*). There is already evidence for non-catalytic regulatory sites in proteasomes (*Schmidtke et al., 2000*; *Kisselev et al., 2002*, *2003*). In particular, *Kisselev et al. (2002)* and *Kisselev et al. (2003)* identified two enhancers of the degradation rate of Suc-FLF-MNA and Suc-LLVY-MNA molecules, but not of Boc-LRR-MCA. We therefore expect that Suc-LLVY-MCA will enhance its own hydrolysis through a positive feedback loop (self-activation). Indeed, the degradation rate of Suc-LLVY-MCA by mouse proteasome progressively accelerates within 90 min (*Figure 1B*), and the degradation rate of the substrate Bz-VGR-MCA increases in the presence of the peptide LLVY (*Figure 3A*)—this is indirect evidence for the existence of feedback. The resulting time course shows an initially low reaction velocity, which then increases over time. This phenomenon is more pronounced at substrate concentration above 40 μM (*Figure 3A*). To test whether the progressive increase of the enhancing effect of LLVY peptide over

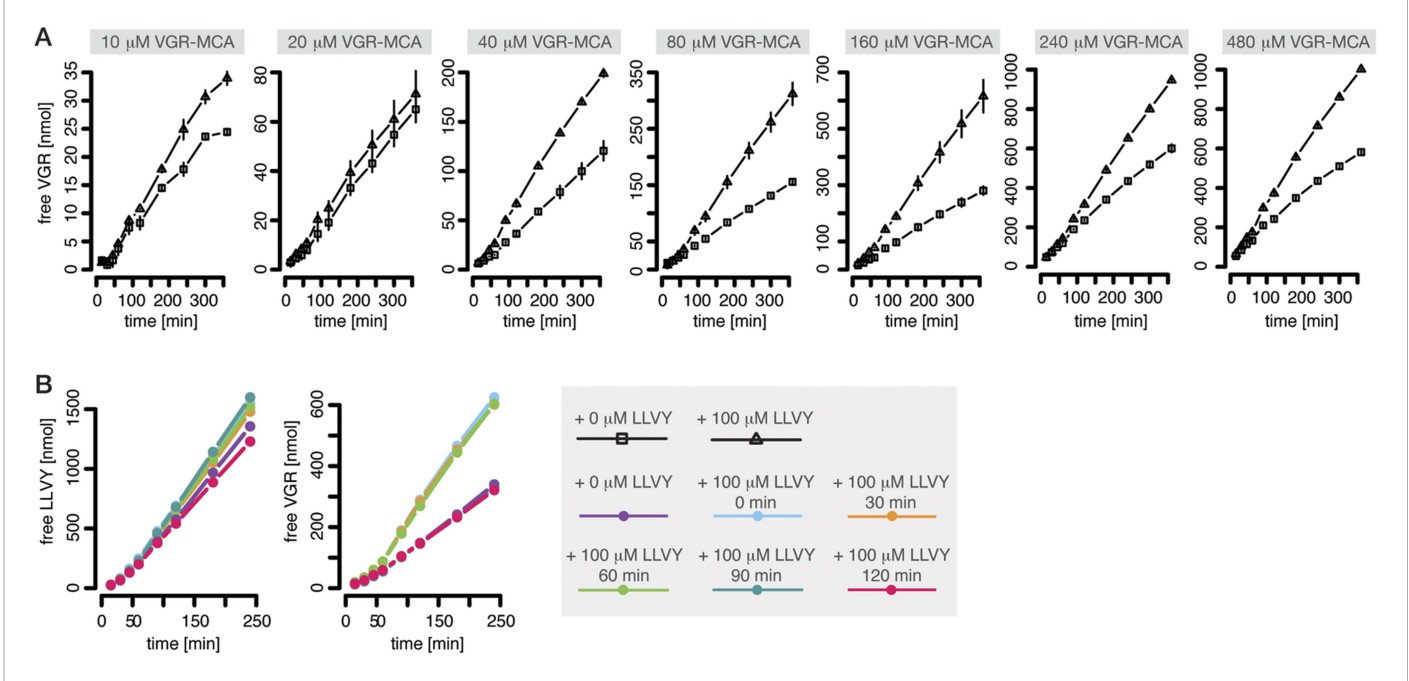

Figure 3. Peptide-mediated enhancement of proteasome activity. (A) Product formation over time from degradation of Bz-VGR-MCA by mouse proteasome in presence or absence of LLVY peptide over time. (B) Product formation from degradation of Suc-LLVY-MCA and Bz-VGR-MCA (100 µM and 200 µM, respectively) after pre-incubation at 37°C of mouse proteasome with LLVY peptide over time.
The following figure supplements are available for figure 3:

**Figure supplement 1.** Compartmentalised model with affinity transport and enhancing regulatory site(s) inside the chamber and substrate inhibitory site (s) outside the chamber and its kinetic parameters.

**Figure supplement 2.** Compartmentalised model with affinity transport and enhancing regulatory site(s) inside the chamber and substrate inhibitory site (s) outside the chamber can simulate the different dynamics of polypeptides cleavage sites as observed in in vitro digestions.

time is due to slow kinetics of the binding to non-catalytic modifier sites we pre-incubate mouse proteasome with LLVY peptide prior to the start of the degradation of the substrates Suc-LLVY-MCA and Bz-VGR-MCA. The enhancing activity of LLVY is maximal for shorter pre-incubation, but the initial acceleration remains the same (*Figure 3B*). Thus the progressive acceleration over time of the two substrates' degradations is not due to slow binding of the LLVY peptide to non-catalytic modifier sites. Instead the data suggest that peptide-bond hydrolysis is mandatory for the enhancing effect by LLVY peptide.

A possible explanation of our observations is that LLVY accumulates slowly inside the proteasome where it binds to the regulatory site and then opens the gate. We can confirm this with the help of Rpt peptides, which are short peptides deriving from subunits of the 19S regulatory complex and known to bind the proteasome α subunit tails and open the gate (*Gillette et al., 2008*). When mouse 20S proteasome digests the substrates Bz-VGR-MCA in presence of Rpt peptides no further enhancement of the degradation rate is seen when the LLVY peptide is added (*Figure 1—figure supplement 4*).

In our model we therefore have to also include regulatory sites to which substrate can bind. Based on the in vitro results we investigate two alternatives: (i) the regulatory site is on the outer surface of the proteasome and therefore accessible to all external peptides (*Figure 2G*); or (ii) the regulatory site is located inside the proteasome chambers and thus only accessible to peptides that are also inside the chambers (*Figure 2H*). In both cases binding to the regulatory site would lead to an increase of the substrate influx and hence to an increase in the model parameters $v_{in}$ and $v_{out}$ by a factor (parameter $X_{enh}$).

We also observe evidence for such a positive feedback loop in the degradation of Z-LLE-MCA, where the speed of the reaction increases more slowly as the substrate concentration increases. (*Figure 1—figure supplement 5*). This substrate inhibition takes place during the first 60 min of the gate opening process. This suggests the existence of a regulatory site, which upon binding of substrate stops the gate from opening. If such a regulatory site were inside the proteasome we should observe the effect only after enough substrate has entered the chambers, that is, only at later time points assuming that the chamber fills only slowly; if, however, the regulatory site were on the proteasome's exterior surface and thus exposed to the initial substrate concentration, inhibition could take place immediately with the initiation of the reaction. Accordingly, we extend the models shown in *Figure 2G* and *Figure 2H* by adding a regulator site outside the proteasome with inhibitory effect on the gate conformation. Such inhibition could occur upon specific binding to a specific site, or upon unspecific binding of peptides to the surface of the proteasome (*Figure 2I,J*).

The last five models (*Figure 2F–J*) all contain the same description of the active site events. We employ the same substrate and product inhibition model as described for the SI model in *Figure 2B* and the active site events of the resulting models are described by the schematic shown in *Figure 1—figure supplement 2*.

We implement all discussed models (*Figure 2A–D* and *Figure 2F–J*) and use Bayesian model selection (*Kirk et al., 2013*) to determine which model best represents our experimental data. First we focus on data generated from mouse proteasome digestion of 80–480 µM Suc-LLVY-MCA (*Liepe et al., 2015*). We perform model selection using our approximate Bayesian computation sequential Monte-Carlo (ABC-SMC) framework (*Toni et al., 2009*). We start by comparing the four non-compartmentalised models (*Figure 4A*). The winning model is then compared to the compartmentalised models in a pairwise manner (*Liepe et al., 2014b*). Here the 'winning' model is compared to the next model, and models are compared in the order of increasing complexity. This iterative scheme (*Figure 4A*) provides a 'best' model, which is then again tested against all other models.

Neither the model without any transport regulation, nor the model with an enhancing regulator site outside the proteasome can explain the substrate inhibition observed in our data (*Figure 4A,B* and *Figure 2—figure supplement 1A,B,D*, dose-response curves). The two models with the enhancing regulatory site inside the chambers can reproduce the increased reaction velocity at early time points observed in the data, but only the model with an inhibiting regulatory site is able to fit simultaneously the substrate inhibition and is therefore able to reproduce the time course data in detail (*Figure 4B* and *Figure 2—figure supplement 1C,E*). This is further confirmed when we use data generated by digestion of 160–640 µM Z-LLE-MCA using mouse proteasome and apply the model selection scheme for the last two models (*Figure 2—figure supplement 2A–C*).

In summary, the integrative analysis suggest that the gate-opening regulatory site proposed previously (*Schmidtke et al., 2000*; *Kisselev et al., 2002*) should be located inside the proteasome chambers. We also find evidence for a transport inhibiting regulatory site located on the proteasome surface. Combining the description of the substrate transport and the substrate hydrolysis at the active site, the resulting model now accounts for the previously described, but never wholly explained, observations: reduction of product generation long before substrate depletion (*Stein et al., 1996*); and substrate inhibition at early and late time points (*Stein et al., 1996*; *Schmidtke et al., 2000*). Furthermore, it explains the reaction velocity increase over time, which results from the spatial organisation of the proteasome.

## Proteasome regulatory experiments are predicted correctly

After model selection we calibrate our mathematical model against the experimental data sets for the degradation of the substrates Suc-LLVY-MCA, Z-LLE-MCA and Bz-VGR-MCA by mouse proteasome. Data and model fits for the substrates are shown in *Figure 3—figure supplement 1A*; we obtain posterior parameter distributions that provide us with confidence intervals for the parameters and allow us to detect potential correlations between parameters. The parameter estimates are shown in *Figure 3—figure supplement 1B* and related to the model in *Figure 3—figure supplement 1C*.

We first test if our kinetic model can qualitatively reproduce published results on proteasome modifier sites (which were not used in model development and calibration). *Kisselev et al. (2002)* showed that proteasomal cleavage of the substrate Boc-LRR-MCA is enhanced by adding Suc-LLVY-MNA (or Suc-FLF-MNA), or by using the mutant Δ3αN proteasome, which has a constitutively fully open gate. Our calibrated model reproduces qualitatively the time course of

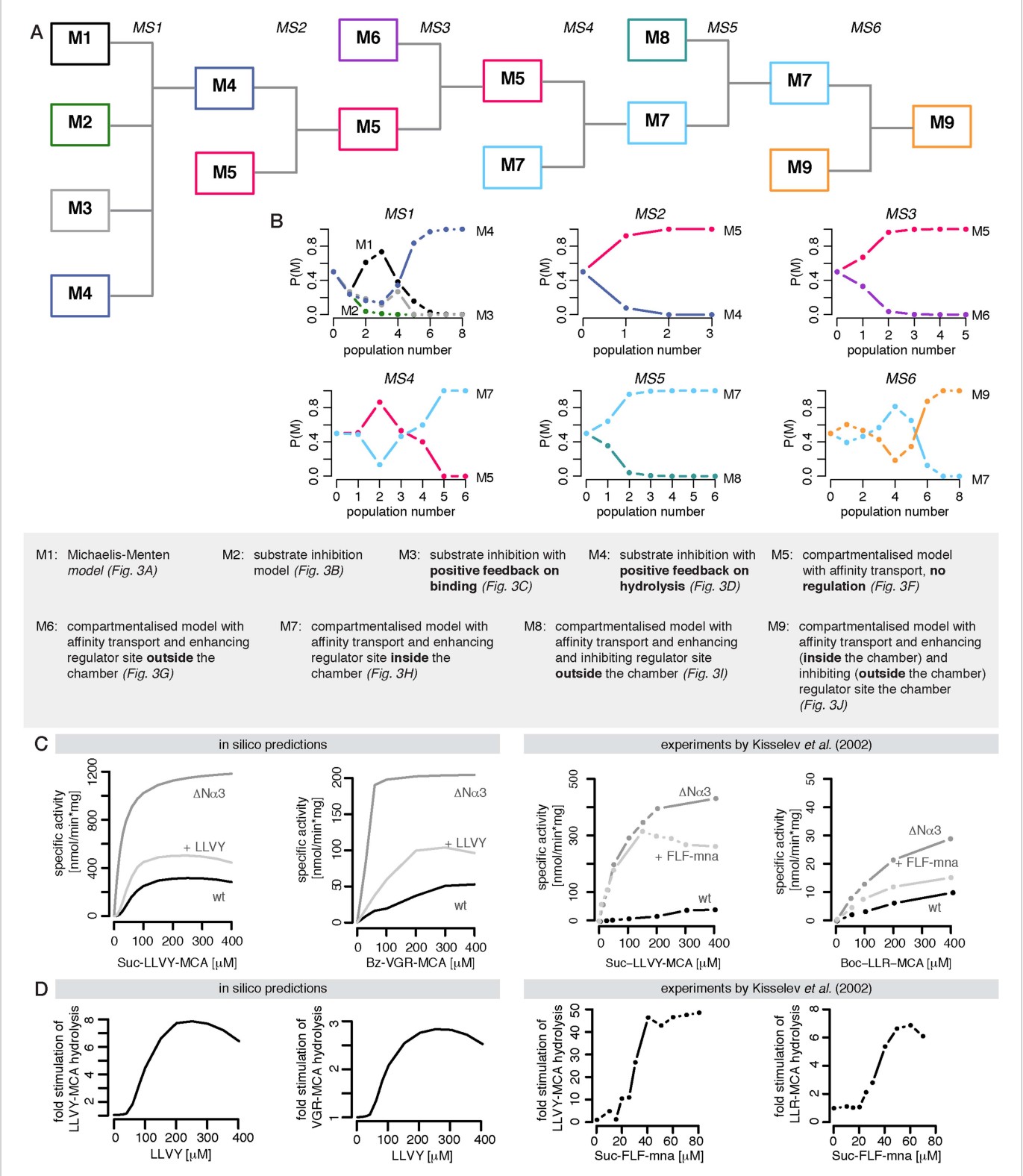

**Figure 4**. Bayesian model selection and model validation. (**A**) A model comparison scheme is applied to identify the best candidate among models represented in *Figure 2F–J*. *MS* stands for model selection. (**B**) The prior model probability is 0.5 for all pairwise model comparisons, and 0.25 for the comparison of models 1–4. The model selection scheme proceeds over SMC populations (*Toni et al., 2009*) each of them returns an updated model probability, until the winning model has a probability of 1 in all comparisons. The colours correspond to (**A**). The winning model is challenged by in silico

*Figure 4. continued on next page*

*Figure 4. Continued*

experiments. (**C**) The posterior parameter distributions inferred from a data set using mouse proteasome (*Figure 3—figure supplement 1A–C*) are used to simulate the mean behaviour of opened-gate mutant (ΔNα3) and the effect of Suc-LLVY-MCA. Simulation of the mutant (ΔNα3) is achieved by increasing the parameters $v_{in}$ and $v_{out}$ 10-fold. The model is extended to simulate the effect of the molecule Suc-LLVY-MCA, parameters are taken from the posterior parameter distribution obtained from digestions of Suc-LLVY-MCA. (**D**) Dose response curves are simulated for the effect of the molecule Suc-LLVY-MCA on the peptide-bond hydrolysis of Suc-LLVY-MCA and Bz-VGR-MCA. In (**C**) and (**D**) the results are qualitatively comparable to the results of the experiment by *Kisselev et al. (2002)*.

The following figure supplement is available for figure 4:

**Figure supplement 1**. Human standard- and immuno-proteasomes vary their cleavage activities over time.

Bz-VGR-MCA and Suc-LLVY-MCA hydrolysis of *Kisselev et al. (2002)* (*Figure 4C*). In the same study the authors also showed dose-response curves for the substrates Suc-LLVY-MCA and Boc-LRR-MCA in dependence of the LLVY peptide concentration; our model also reproduces the results for the hydrolysis of the substrates Suc-LLVY-MCA, Z-LLE-MCA and Bz-VGR-MCA (*Figure 4D*). To note, in the study by *Kisselev et al. (2002)* the authors used proteasome purified from other species rather than mouse and different short fluorogenic peptides (Suc-FLF-MNA instead of Suc-LLVY-MCA and Boc-LLR-MCA instead of Bz-VGR-MCA). These differences in the experimental setup explain why we obtain only qualitative agreement between the published data and our in silico predictions.

## Modelling polypeptide degradation

The hydrolysis of oligo- and polypeptides has previously been modelled phenomenologically (*Liepe et al., 2014a*). Our model, calibrated against degradation data of short peptides, can serve as starting point for modelling polypeptide degradation; but many possible substrate cleavage sites will make the investigation of polypeptide degradation computationally and experimentally more challenging.

As a first step we are interested in whether our fully parameterised model can explain the altered cleavage site usage over time (*Figure 1C,D*). We extend the model in order to describe the hydrolysis of a hypothetical peptide with two cleavage sites. A schematic of the substrate and possible resulting products is shown in *Figure 3—figure supplement 2A*. To reduce computational complexity we model the allosteric regulation of the active sites in a simplified fashion (parameters: $K_{iS}$, $K_{iP}$, $n_i$, $n_a$, $\alpha$, $\beta$); all other reaction steps and the non-catalytic regulatory mechanisms are as above. We assume that the parameters related to the peptide-bond hydrolysis at the active sites are the same for substrate and all resulting products, but the parameters related to peptide transport are substrate dependent. This allows us to test whether the cleavage site usage variation over time can be explained by transport properties of the substrate and products.

In our framework we detect all possible substrate cleavage site behaviours that we observe by digesting representative polypeptides (*Figure 1C,D* and *Figure 3—figure supplement 2B–E*).

## Comparing human s- and i-proteasome dynamics

### S- and i-proteasome have different kinetic parameters

In the last decade potentially different activities of s- and i-proteasomes have been discussed, at times controversially, although recent results (*Mishto et al., 2014*) suggest that such differences are not of qualitative but quantitative nature. Like we have observed for the mouse proteasome, human 20S proteasome purified from T2 (s-proteasome) and LcL (i-proteasome) cell lines shows degradation dynamics of short fluorogenic peptide that vary over time (*Figure 4—figure supplement 1A*). This is not due to variation of proteasome functionality as pre-incubation of the proteasome without substrate at 37°C for 18 hr does not alter its activity (*Figure 4—figure supplement 1B*), in agreement with what is observed for mouse proteasome (*Figure 1—figure supplement 1A*). Similarly, the substrate cleavage preferences within the $LLO_{291–317}$ polypeptide by s- and i-proteasomes vary over time, and not because of product re-entry. We observe modification of the cleavage site usage already at early time points (*Figure 4—figure supplement 1C*). However, a slight reduction in the average peptide products length, as would be expected to result from further peptide product fragmentation, is evident only at late time points and with less than the 50% of the substrate still intact

(*Figure 4—figure supplement 1D*). This observation is in agreement with an analysis of the production efficiency of peptides derived from the tested polypeptide substrate: these vary already at early time points and show marked differences between s- and i-proteasomes (*Mishto et al., 2014*).

In order to understand the origin of the kinetic differences between s- and i-proteasomes we calibrate our mechanistic model against new time course data from s- and i-proteasomes. We then compare the resulting posterior parameter distributions to determine which kinetic parameters differ between s- and i-proteasomes. Finally, we make in silico predictions about the different effects of the Rpt peptides on the substrate hydrolysis by s- and i-proteasomes, which we then validate experimentally.

We use measurements for six different initial concentrations (from 20 to 640 µM) of the substrates Suc-LLVY-MCA, Z-LLE-MCA and Bz-VGR-MCA, using 0.5 µg of the s- and i-proteasomes. *Figure 5—figure supplement 1* shows that the model is able to reproduce all experimental data sets. Due to the presence of the catalytic immuno-subunits in i-proteasomes we expected to observe differences in the parameters related to the active sites. Suc-LLVY-MCA is mainly hydrolysed by the β5/β5i subunits (*Mishto et al., 2014*), and we observe a higher active site affinity ($K_{aS}$) with a lower Hill coefficient ($n_a$) in i- than in s-proteasomes; similarly i-proteasomes have a lower inhibitory site affinity ($K_{iS}$) with a higher Hill coefficient ($n_i$) compared to s-proteasomes. Furthermore, the hydrolysis rate ($k_p$) is higher in i-proteasomes (*Figure 5A*).

The posterior parameter distribution of Z-LLE-MCA (mainly cleaved by β1/β1i active sites) (*Kisselev et al., 2003*) also shows differences in active site parameters, and for i-proteasomes we observe lower affinity to the active site ($K_{aS}$) and a lower Hill coefficient regarding binding to the inhibitory site ($n_i$). We find no evidence for differences in parameters related to the active site for Bz-VGR-MCA (mainly cleaved by β2/β2i active sites).

Comparing posterior parameter distributions for s- and i-proteasomes reveals differences in parameters related to peptide transport and transport regulation for all three substrates (*Figure 5B*). All three substrates have altered affinities to the gate ($k_{off}/k_{on}$: i-proteasome lower than s-proteasome for Suc-LLVY-MCA and Z-LLE-MCA but higher for Bz-VGR-MCA) and altered influx and efflux rates ($v_{in}$, $v_{out}$: i-proteasome higher for Suc-LLVY-MCA and Bz-VGR-MCA but lower for Z-LLE-MCA). Finally, affinity to the enhancing regulatory site inside the proteasome chambers ($R_{off}/R_{on}$) is higher in i-proteasome for all three substrates. By contrast, we find no evidence for differences in $X_{enh}$ and $\tau$. Furthermore, the marginal posterior parameter distributions for the maximal capacity $C$ do not differ between s- and i-proteasomes, which was expected because both isoforms have the same proteasomal cavity volume.

In summary, the presence of the immuno-subunits in the assembled proteasome influences not only active site parameters but also the parameters that regulate substrate transport.

## Peptide transport is the main limiting step in human s- and i-proteasome

To further understand differences between s- and i-proteasome and how substrate transport- and hydrolysis steps affect overall substrate degradation, we perform sensitivity analysis based on the posterior parameter distribution. Sensitivity coefficients inform us about the parameters that determine the overall dynamics (*Shen, 1999*; *Wu et al., 2008*). Since we are interested in the rate limiting steps of the proteolysis, we need to determine the reaction that has the strongest potential to increase product formation. We test the influence of the gate affinity, peptide influx, hydrolysis, peptide translocation and peptide efflux as well as initial gate size (simultaneous change of influx and efflux) on the product formation. We compute the fold-change in product formation when increasing one of these reactions (*Figure 5C*), and perform this analysis in silico using 320 µM substrate (the same analysis using 80 µM can be found in *Supplementary file 1*). The reaction inducing the strongest fold change in product formation is the rate-limiting step.

For Suc-LLVY-MCA we find that the initial gate size has the highest impact. Increasing the hydrolysis also increases the product formation, but to a lesser extent. This is observed for s- and i-proteasomes, and is valid independently of substrate concentration (*Figure 6—figure supplement 1A*); similar results are found for Bz-VGR-MCA. The degradation of Z-LLE-MCA is limited by the efficiency of the hydrolysis, which induces the strongest increase in product formation. None of the peptide transport related reactions can increase the overall proteasome activity. When using 80 µM Z-LLE-MCA the hydrolysis is still the main rate limiting step, but the gate size has a similar impact (*Figure 6—figure supplement 1A*). Reasons for the observed differences might lie in the physical and

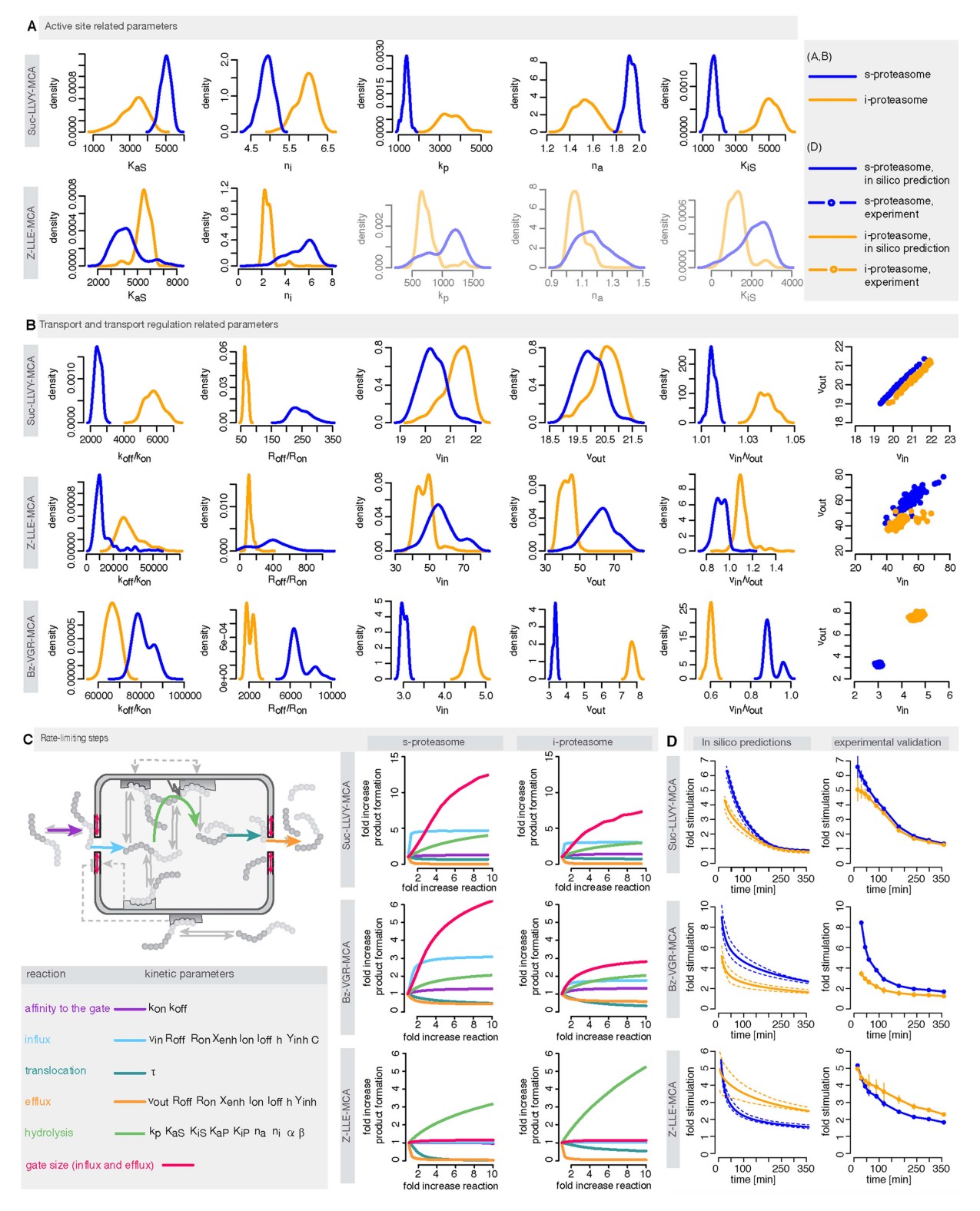

**Figure 5.** Human s- vs i-proteasomes. (**A**) Marginal posterior parameter distributions for active site related parameters that differ between s- and i-proteasomes. No evidence for differences related to Bz-VGR-MCA was detected. (**B**) Marginal posterior parameter distributions for transport related parameters that differ between s- and i-proteasomes. (**C**) Analysis of rate limiting steps in s- and i-proteasome. Shown is the fold increase of product formation upon increase of a specific reaction. Substrate concentration is 320 μM, measurement is taken after 60 min reaction. (**D**) In silico predictions for

*Figure 5. continued on next page*

*Figure 5. Continued*

fold stimulation of substrate hydrolysis in presence of Rpt peptides and experimental validation. A final concentration of 40 µM Rpt peptides was added to the standard experimental setup described in 'Materials and methods'. Dashed lines indicate 5%- and 95%-iles of the predictions.

The following figure supplement is available for figure 5:

**Figure supplement 1**. Fitting of experimental data using the compartmentalised models with affinity transport en- hancing regulatory site(s) inside the chamber and substrate inhibitory site(s).

chemical properties of the investigated substrates, which differ not only in their volumes, but also in their partial charges.

Overall peptide transport is the rate limiting step in proteasomal degradation. Our model shows how even a subtle difference in one of the transport parameters can result in strongly altered product formation kinetics. Differences in kinetic parameters between s- and i-proteasomes furthermore result in different chamber-filling kinetics (*Figure 6*), which are also reflected by the differences in rate limiting steps. Equivalent results are also obtained for the mouse proteasome (*Box 1* and *Figure 6—figure supplement 1B–D*).

Since substrate-characteristic peptide transport appears to shape the kinetics of (20S) proteasome-mediated peptide degradation, this process should naturally be the rate limiting-step of the overall reaction; furthermore it seems to vary systematically between s- and i-proteasomes.

To test the robustness of our posterior parameter distributions we alter specific steps of the proteolytic process. First, we predict in silico the kinetic effect of the Rpt peptides—which open the gate by binding the α subunit tails (*Gillette et al., 2008*)—on the hydrolysis rate of s- and i-proteasomes over time. According to our simulation the enhancing effect of Rpt should be strongest at early time points and decrease over time (*Figure 5D*). We also predict that at early time points the effect of Rpt will be larger in s-proteasome when hydrolysing Suc-LLVY-MCA than in i-proteasomes, although over time the enhancing effects are the same. A similar effect is predicted for proteasomal hydrolysis of the Bz-VGR-MCA substrate in presence of Rpt peptides, although s-proteasome is predicted to remain higher than i-proteasome also at later time points (*Figure 5D*). Third, we predict that both proteasome types will initially have the same enhancing effect while digesting Z-LLE-MCA substrate, but that this effect becomes stronger over time for i-proteasomes. The in vitro experiments verify these three nontrivial in silico predictions. The small quantitative deviations can be explained by the use of a simplified model for the functioning of Rpt peptides (multiplication of the parameter $v_{in}$ by a constant factor), which is expected to be more complex in reality. But this comparison validates that transport is indeed the rate limiting step (*Figure 5A–C*).

## Discussion

Using tightly integrated experimental and computational modelling analyses, recent advances in our understanding of proteasome structure and function, and previous attempts at modelling proteasome dynamics (*Liepe et al., 2014a*), we elaborate the first comprehensive mathematical model that is able to describe the regulatory components of 20S proteasomes; and the complex interactions between substrate/product transport and substrate hydrolysis over time for representative peptides. Modelling helps us to understand how the 20S proteasome catalyzes the degradation of specific proteins in cells (*Pickering and Davies, 2012*; *Ben-Nissan and Sharon, 2014*; *Höhn and Grune, 2014*; *Fabre et al., 2015*); it also forms the basis for future studies of the main active forms of proteasome in cells, for example, when it is bound to the regulatory complexes 19S and PA28 (*Fabre et al., 2015*). Indeed, in 19S or PA28 single-capped proteasomes all steps considered in our model are still present, including the regulation of the gate by non-catalytic modifier site(s) and the gate binding affinity since one of the proteasome gates is not bound to the regulatory complexes and will therefore be regulated as described here.

We observe similar proteasome dynamics over time when using purified 20S proteasome as well as protein homogenates, where 20S proteasomes are surrounded by several regulatory molecules. It suggests that with the present model we describe a core of proteasome features that could reflect proteasome dynamics also in more complex scenarios such as might exist inside the cellular

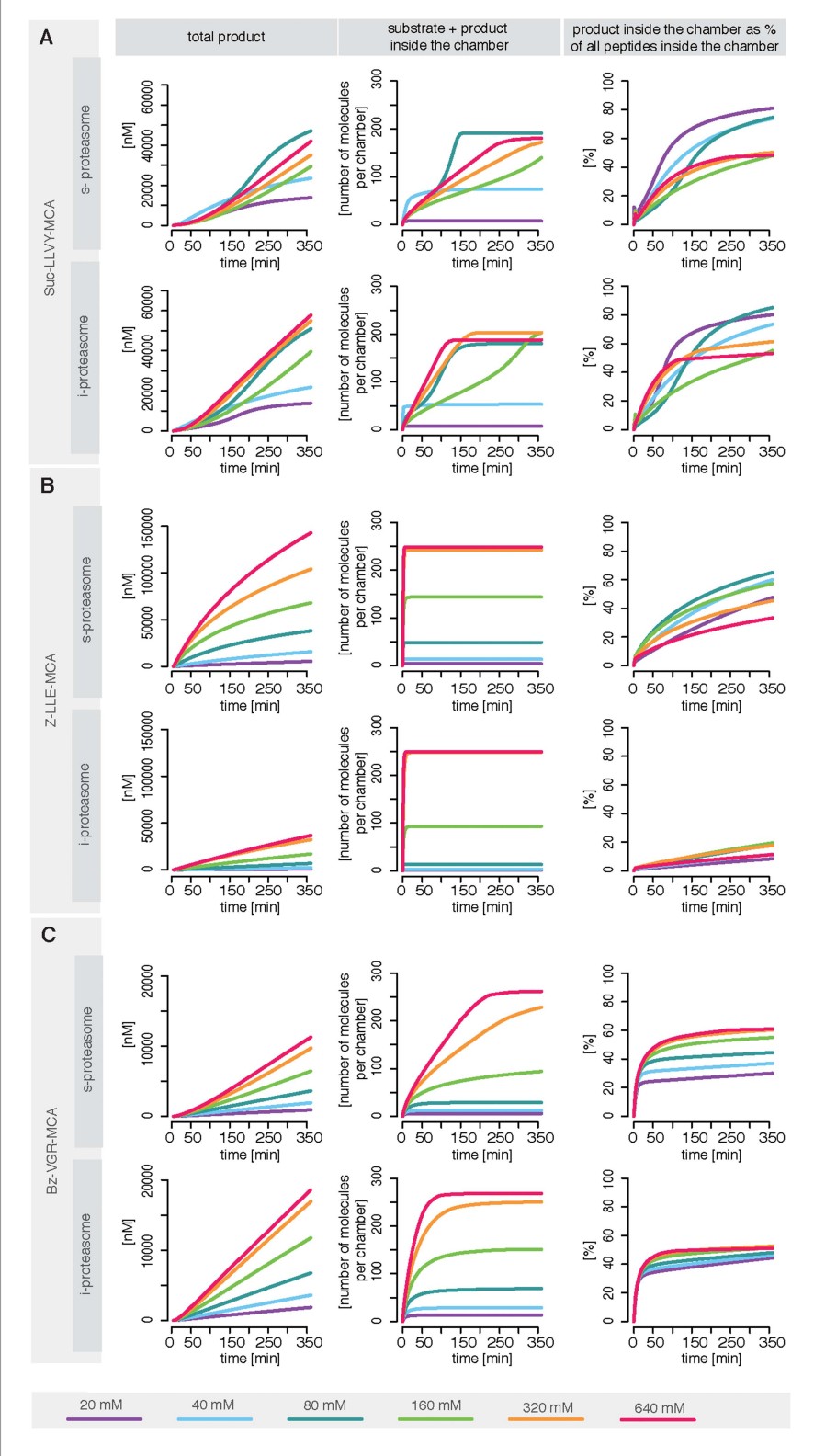

**Figure 6.** Rate limiting steps of human proteasome activity. The mean of in silico predictions (coloured lines) is plotted over time for the degradation of the substrates Suc-LLVY-MCA (**A**), Z-LLE-MCA (**B**) and Bz-VGR-MCA (**C**) with varying initial substrate concentrations using human s- and i-proteasomes, respectively. The inferred posterior parameter distributions of each substrate were used to simulate the number of peptide molecules (product and

*Figure 6. continued on next page*

*Figure 6. Continued*

substrate) and the relative amount of product vs total amount of peptides inside the chambers over time.

The following figure supplement is available for figure 6:

**Figure supplement 1**. Rate limiting steps of proteasome peptide degradation.

environment. Our model therefore also provides an interpretative framework for general studies of proteasome action, keeping in mind that substrate-specific effects need to be accounted for (*Mishto et al., 2014*). Simple (e.g., MM-type) models, which have been largely used in analysing proteasome functionality so far, are clearly not capable of describing proteasome dynamics; this is becoming apparent since time-course data is much less forgiving and harder to fit to than for example, dose-response curves. The 20S proteasome is a dynamic enzyme and its catalytic activity varies over time in a way that can only be understood if the structural characteristics of the proteasome are taken into

## Box 1. Peptide transport is one of the rate-limiting steps of the substrate degradation by mouse 20S proteasome.

To understand the details of the proteolytic activity we use our fitted model and predict the proteasome dynamics for different substrate concentrations (*Figure 6—figure supplement 1*). Even though experimentally we can only observe the total amount of product produced over time (grey dots in *Figure 6—figure supplement 1*), the kinetic model now provides information about the separate steps involved in the substrate degradation. In *Figure 6—figure supplement 1* we plot the total product concentration (*Figure 6—figure supplement 1B*), the amount of substrate and product inside the proteasome chamber over time (*Figure 6—figure supplement 1C*). For all three substrates we observe that with increasing substrate concentration the proteasome chamber fills up faster with peptides. For Z-LLE-MCA we observe a more rapid filling compared to Suc LLVY-MCA and Bz-VGR-MCA, where the filling lasts approximately 6 hr. When plotting the amount of product inside the chamber relative to the total amount of peptides in the chamber, we find that Suc-LLVY-MCA and Bz-VGR-MCA are cleaved immediately once inside the chamber (90% of peptides are products) (*Figure 6—figure supplement 1D*). This shows that, for the substrates Suc-LLVY-MCA and Bz-VGR-MCA, the transport inside the chamber, rather than the binding to the catalytic site and the peptide-bond hydrolysis, regulates how fast these substrates are degraded. The posterior parameter distributions show that Z-LLE-MCA substrate is transported faster than Suc-LLVY-MCA and Bz-VGR-MCA (see *Figure 3—figure supplement 1B*, vin and $v_{out}$). This results in the rapid accumulation of Z-LLE-MCA molecules inside the proteasome chamber. Even though the catalysis rates ($k_p$) of the three substrates are the same, Z-LLE-MCA hydrolysis is strongly influenced by both product and substrate inhibition (this is indicated by the very small values for the parameter β). Because of that Z-LLE-MCA will be cleaved less efficiently (30–50%) than the other substrates (*Figure 6—figure supplement 1D*). Here modeling can elucidate those processes that are only measurable indirectly.

In summary, in light of our experiments and models the substrate transport appears to be the most important factor for controlling how fast short fluorogenic substrates are degraded by mouse 20S proteasome, and therefore it is the rate-limiting step of their hydrolysis. The hydrolysis of substrates that accumulate easily inside the proteasome chamber can be additionally influenced by strong substrate and product inhibition effects resulting in less efficient substrate degradation.

account. Kinetics change over longer time scales than the structural dynamics described by *Osmulski et al. (2009)* and they are influenced by the interactions of substrates and products with both proteolytic and regulatory sites. Our analysis suggests that the cavity of 20S proteasomes is progressively filled by peptides that can further facilitate other molecules to enter the chambers until the effective cavity volume has been almost completely filled. In fact, the estimated numbers of substrate and product molecules (*C*) that are inside the proteasome cavity are close to the possible maxima (*Box 2*). Peptide accumulation over time leads to activation of non-catalytic modifier sites, which our analysis shows to be located inside the proteasome cavity.

Substrate transport through the gate and cavities must therefore be a key factor in the regulation of the proteolytic process, despite the fact that it had been largely neglected so far in the quantitative analysis of proteasome dynamics (*Liepe et al., 2014a*). The integration of data into our mathematical model clearly shows that peptide transport is the major rate limiting step in the degradation of short fluorogenic substrates: all models/hypotheses that do not account for this are soundly rejected. Hydrolysis is only rate limiting for the degradation of Z-LLE-MCA. We stress that for the short fluorogenic substrates the rate limiting steps are conserved across the different proteasomes (mouse proteasome and human s- and i-proteasomes). This in turn suggests that the interspecies homology of the proteasome structure results in preserving transport characteristics, although subtle structural variations can quantitatively modify the transport, as observed by comparing s- and i-proteasomes.

Variation of the transport efficiency typically leads to quantitative changes in the degradation rates of short fluorogenic substrates. For more complex substrates, for example, polypeptides, transport variation could change substrate cleavage-site usage and thus result in the generation of specific peptide products. If these peptides are MHC class I-restricted epitopes, substantial variations in their amount could strongly affect the cell-mediated immune response. Indeed, as we have recently shown, significant decreases of antigenic peptide amount produced by proteasome could lead to a presentation onto the MHC class I molecules that is so diminished to be not able to trigger CD8+ T cell activation (*Mishto et al., 2014*). According to our model, modifications of the proteasome gate upon PA28 binding will for instance affect the substrate cleavage-site usage and lead to the observed alterations of MHC class I-restricted epitope repertoire (*Cascio, 2014*).

## Box 2. Filling kinetics of the proteasome inner cavity over time.

A key part of the peptide transport dynamics is the maximum capacity (C) of the two proteasome ante-chambers and the main chamber. C describes the maximal number of molecules of a given substrate can be allocated inside the proteasome chamber at the same time. C is dependent on the volume of the proteasome chamber and on the volume of the substrate molecules. For this reason C is expected to be substrate specific. The model estimates of C for the three tested substrates are in agreement with values predicted by computing the maximal number of substrates that could be located inside the proteasome chamber. The estimated volumes for Suc-LLVY-MCA, Z-LLE-MCA and Bz-VGR-MCA are approx. 936 $A^3$, 741 $A^3$ and 606 $A^3$, respectively (with an MCA group of 200 $A^3$). This results in a ratio of 1.25 for Suc-LLVY-MCA:Z-LLE-MCA (model estimate: 1.24), 1.36 for Suc-LLVY-MCA:Bz-VGR-MCA (model estimate: 1.46) and 1.08 for Z-LLE-MCA:Bz-VGR-MCA (model estimate: 1.12). The exact number of molecules inside the chamber depends on the maximum possible density, but is approximately estimated to be 250, 310 and 337 for Suc-LLVY-MCA, Z-LLE-MCA and Bz-VGR-MCA, respectively, which is in the same order of magnitude as the model estimates (200, 249 and 293). Note, computed values are the maximal possible amount of molecules that could be packed inside the chamber. However, under physiological conditions the actual number of molecules that can be filled in the proteasome cavity has to be smaller, which is reflected in our model estimates.

Differences in gate and inner channel features are in part responsible for the different degradation rates of the short fluorogenic peptides by human s- and i-proteasomes although they diverge also in the activity of the catalytic subunits themselves. Differences in the gate between the two proteasome isoforms are supported also by *Fabre et al. (2015)*, who identified preferential binding between the α subunits of s- and i-proteasomes and specific regulatory complexes. Such differences can become dominant driving forces. For instance, the degradation of the substrate Bz-VGR-MCA is not influenced by differences in the active site-related parameters—as is correctly predicted by our model. This also agrees with comparisons of the crystallographic structures of the mouse s- and i-proteasomes, which identify only minor variations between the β2 and β2i catalytic pockets (*Huber et al., 2012*). By contrast, the catalytic sites of s-proteasome have stronger binding affinity for the substrate Z-LLE-MCA than the i-proteasome, which could be explained by the structural differences between β1 and β1i catalytic pockets described by *Huber et al. (2012)*. For the degradation of the substrate Suc-LLVY-MCA our model predicts that the i-proteasome has higher substrate-binding affinity, higher hydrolysis rate, and lower cooperativity than the s-proteasome. This prediction is corroborated by the recent study of *Arciniega et al. (2014)*, who showed that the β5i pocket is more prone to bind and process substrates than the β5 pocket; however, only s-proteasome is able to modify its conformation upon substrate binding to the β5 subunit (*Arciniega et al., 2014*), which is also correctly captured by our model as increased cooperativity ($n_i$).

The outcomes of our integrative analysis suggest that the i-proteasome also has a higher affinity for the enhancing non-catalytic modifier site activated by all three substrates; given that our model consistently predicts this site to be located at the inner surface, it should now become possible to determine the precise location of the enhancer site using structural techniques.

Our model is a necessary and non-trivial step towards understanding protein degradation by proteasomes. A full-length protein is a much more complicated substrate and it can interact with the different regulatory and catalytic sites in a myriad of ways. In any such analysis, however, the model developed here can aid the experimental set-up (*Liepe et al., 2013*; *Sunnaker et al., 2013*) and the interpretation of experimental data. What is already clear is that the complex interactions between a suitable complicated substrate and the proteasomal machinery will substantially shape differences in degradation rates between proteins; as the protein degradation rate is known to influence for example, noise in signal transduction, a way to understand and rationally interfere with this process is of obvious importance.

# Materials and methods

## Experimental procedures

### 20S proteasome purification and protein homogenates
20S proteasomes from LcL and T2 cells and mouse liver are purified as previously described (*Mishto et al., 2014*). Cell protein homogenates are extracted from T2 cells as previously described (*Mishto et al., 2006*).

### In vitro digestion of synthetic polypeptides and short fluorogenic peptides
Synthetic polypeptides or short fluorogenic peptides are digested by purified 20S proteasomes or cell protein homogenates in 100 µl TEAD buffer (Tris 20 mM, EDTA 1 mM, NaN$_3$ 1 mM, DTT 1 mM, pH 7.2) over time at 37°C as previously described (*Mishto et al., 2014*).

### Peptide synthesis and quantitation
Peptides gp100$_{35-57}$ (VSRQLRTKAWNRQLYPEWTEAQR), LLO$_{291-317}$ (AYISSVAYGRQVYLKLSTNSHS TKVKA), Rpt2 (GTPEGLYL) and Rpt5 (KKKANLQYYA) are synthesized using Fmoc solid phase chemistry as previously described (*Mishto et al., 2012*). Quantification of produced peptides—both cleavage and spliced products—and computation of the substrate site-specific cleavage strength (SCS) are carried out by applying QME method to the LC-MS analyses (*Mishto et al., 2012*).

## Mathematical model
The final model shown in *Figure 2J* contains a set of ordinary differential equations as follows:

$$\frac{dS_{out}}{dt} = -S_{out}G_1 k_{on} + [G_1 S_{out}]k_{off} + [G_2 S]transport_{out} - hS_{out}^h I_{free}I_{on} + h[IS]I_{off}$$

$$\frac{dG_1}{dt} = -(S_{out} + P_{out})G_1 k_{on} + ([G_1 S_{out}] + [G_1 P_{out}])(k_{off} + transport_{in})$$

$$\frac{dP_{out}}{dt} = -P_{out}G_1 k_{on} + [G_1 P_{out}]k_{off} + [G_2 P]transport_{out} - hP_{out}^h I_{free}I_{on} + h[IP]I_{off}$$

$$\frac{d[G_1 S_{out}]}{dt} = S_{out}G_1 k_{on} - [G_1 S_{out}](k_{off} + transport_{in})$$

$$\frac{d[G_1 P_{out}]}{dt} = P_{out}G_1 k_{on} - [G_1 P_{out}](k_{off} + transport_{in})$$

$$\frac{dS}{dt} = [G_1 S_{out}]transport_{in} - \tau\frac{SG_2}{E_0} - v_{hydr} - R_{on}\frac{SE_{reg}}{E_0} + R_{off}[E_{reg}S]$$

$$\frac{dP}{dt} = [G_1 P_{out}]transport_{in} - \tau\frac{PG_2}{E_0} + v_{hydr} - R_{on}\frac{PE_{reg}}{E_0} + R_{off}[E_{reg}P]$$

$$\frac{d[G_2 S]}{dt} = \tau\frac{G_2 S}{E_0} - [G_2 S]transport_{out}$$

$$\frac{d[G_2]}{dt} = \tau\frac{G_2(S+P)}{E_0} - ([G_2 S] + [G_2 P])transport_{out}$$

$$\frac{d[G_2 P]}{dt} = \tau\frac{G_2 P}{E_0} - [G_2 P]transport_{out}$$

$$\frac{d[E_{reg}]}{dt} = -\frac{R_{on}E_{reg}}{E_0}(S+P) + R_{off}([E_{reg}S] + [E_{reg}P])$$

$$\frac{d[E_{reg}S]}{dt} = \frac{R_{on}E_{reg}}{E_0}S - R_{off}[E_{reg}S]$$

$$\frac{d[E_{reg}P]}{dt} = \frac{R_{on}E_{reg}}{E_0}P - R_{off}[E_{reg}P]$$

$$\frac{dI_{free}}{dt} = -(S_{out}^h + P_{out}^h)I_{free}I_{on} + I_{off}([IS] + [IP])$$

$$\frac{d[IS]}{dt} = S_{out}^h I_{free}I_{on} + I_{off}[IS]$$

$$\frac{d[IP]}{dt} = P_{out}^h I_{free}I_{on} + I_{off}[IP],$$

with

$$transport_{in} = v_{in}\frac{1 + X_{enh}\dfrac{[E_{reg}S] + [E_{reg}P]}{E_0}}{1 + Y_{inh}\dfrac{[IS] + [IP]}{I_0}}tanh(E_0 C - S - P)$$

$$transport_{out} = v_{out}\frac{1 + X_{enh}\dfrac{[E_{reg}S] + [E_{reg}P]}{E_0}}{1 + Y_{inh}\dfrac{[IS] + [IP]}{I_0}}$$

$$v_{hydr} = \frac{n_a k_p E_0 S}{xK_{aS}}\left(1 + \frac{\beta S^{n_i}}{\alpha K_{iS}} + \frac{\beta P^{n_i}}{\alpha K_{iP}}\right)$$

$$x = 1 + \frac{S^{n_a}}{K_{aS}} + \frac{S^{n_i}}{K_{iS}} + \frac{P^{n_a}}{K_{aP}} + \frac{P^{n_i}}{K_{iP}} + \frac{S^{n_a + n_i}}{\alpha K_{aS}K_{iS}} + \frac{P^{n_a + n_i}}{\alpha K_{aP}K_{iP}} + \frac{S^{n_a}P^{n_i}}{\alpha K_{aS}K_{iP}} + \frac{S^{n_i}P^{n_a}}{\alpha K_{iS}K_{aP}}.$$

A full list of parameters and model species is given in *Figure 2—source data 2, 3*. Details about the 'Materials and methods', mathematical analysis and other tested models can be found in *Supplementary file 1*. The model is also provided in SBML format (generated with the SBML editor from MINRES Technologies) in *Supplementary file 1*.

## Acknowledgements

The project was in part granted by AICE FIRE Onlus Emilia Romagna to MM, by BIH (Berlin Institute of Health, CRG1-TP1) and Einstein Stiftung Berlin (A2013-174) to PMK, by NC3Rs through a David Sainsbury Fellowship to JL (NC/K001949/1), by BBSRC (BB/G007934/1), HFSP (RGP0061/2011), The Leverhulme Trust (F/07058/BP) and the Royal Society through a Wolfson Research Merit Award to MPHS. We thank Christin Keller and Kathrin Textoris-Taube for technical assistance, Prof. Michael Groll for helpful discussions, Prof. Alexei F Kisselev and Prof. Alfred L Goldberg for allowing the use of their published data. We thank the development team of MINRES technologies for providing us with their SBML editor and for technical support in generating the SBML model files.

## Additional information

### Funding

| Funder | Grant reference | Author |
| --- | --- | --- |
| Biotechnology and Biological Sciences Research Council (BBSRC) | BB/G007934/1 | Michael PH Stumpf |
| Berlin Institute of Health | CRG1-TP1 | Peter M Kloetzel |
| Einstein Stiftung Berlin | A2013-174 | Peter M Kloetzel |
| National Centre for the Replacement, Refinement and Reduction of Animals in Research (NC3Rs) via a David Sainsbury Fellowship | NC/K001949/1 | Juliane Liepe |
| Human Frontier Science Program (HFSP) | RGP0061/2011 | Michael PH Stumpf |
| Leverhulme Trust | F/07058/BP | Michael PH Stumpf |
| The Royal Society | Wolfson Research Merit Award | Michael PH Stumpf |
| AICE FIRE Onlus Emilia Romagna | | Michele Mishto |

The funders had no role in study design, data collection and interpretation, or the decision to submit the work for publication.

### Author contributions

JL, MPHS, Conception and design, Analysis and interpretation of data, Drafting or revising the article; H-GH, Conception and design, Analysis and interpretation of data; EB, Performed the experiments; PMK, Analysis and interpretation of data, Drafting or revising the article; MM, Conception and design, Acquisition of data, Analysis and interpretation of data, Drafting or revising the article

### Author ORCIDs

Michael PH Stumpf, http://orcid.org/0000-0002-3577-1222

## Additional files

### Supplementary file

• Supplementary file 1. Quantitative time-resolved analysis reveals intricate, differential regulation of standard and immune-proteasomes.

## Major dataset

The following dataset was generated:

| Author(s) | Year | Dataset title | Dataset ID and/or URL | Database, license, and accessibility information |
|---|---|---|---|---|
| Liepe J, Holzhütter H, Bellavista E, Kloetzel PM, Stumpf MPH, Mishto M | 2015 | Data from: Quantitative time-resolved analysis reveals intricate, differential regulation of standard- and immuno-proteasomes | http://dx.doi.org/10.5061/dryad.nk453 | Available at Dryad Digital Repository under a CC0 Public Domain Dedication. |

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
