## [Decision Letter]

Thank you for submitting your work entitled “Quantitative time-resolved analysis reveals intricate, differential regulation of standard- and immuno-proteasomes” for peer review at *eLife*. Your submission has been favorably evaluated by John Kuriyan (Senior editor), a Reviewing editor, and three reviewers, one of whom, Shinya Kuroda, has agreed to reveal his identity.

The reviewers have discussed the reviews with one another and the Reviewing editor has drafted this decision to help you prepare a revised submission.

The authors analyzed the mechanism of proteasomal protein degradation by taking advantage of integration of both experimental and mathematical modelling approaches. They developed the alternative mechanistic models of proteasome proteolysis by comparison with experimental data. In particular, they sequentially added mechanistic components on the simple model one-by-one based on both literatures and discrepancies with experimental observation. This is quite reasonable and careful way to develop the alternative models. They performed the statistical model selection using ABC-SMC, and found that the compartmentalised model with affinity transport and enhancing (inside) and inhibiting (outside) regulatory sites is the most plausible. Using the selected model, they further analyzed two distinct types of proteasomes; s- and i-proteasomes. They characterized the quantitative difference between the two proteaomes by posterior parameters distributions, and found that the gate size is a rate-limiting step for Suc-LLVY-MCVA and Bz-VGR-MCA, while hydrolysis for Z-LLE-MCA. Such distinct characteristics were also confirmed by experiments.

This is very elegant study for selecting mechanistic models using both experiments and statistical approaches. The pipeline of model development is clear and well-designed. Furthermore, they gained the new insight of distinct quantitative difference between s- and i-proteasomes, which is intuitively impossible to predict and further validated by experiments. Thus, this study is one of the remarkable and successful examples of analyzing complex biochemical processes using an iterative and tightly integrated experimental and modelling approach.

Essential revisions:

The manuscript's biological conclusions predominantly rely on the interpretation of dynamic experimental data (which appear to be of good quality) with dynamic mathematical models, including Bayesian model selection for the inference of potential mechanisms. While therefore being central to the work, the modeling component raises the following concerns:

1) Model descriptions: The presentation in [Supplementary-material SD4-data], section 2 is partially incomplete and it contains errors that may impact the overall conclusions (for detailed assessment, the authors should provide models and experimental data in appropriate numerical exchange formats). Major missing data are the initial conditions for all models/simulations, e.g. assumptions on numbers of gates (G_i_) cannot be assessed/compared between simulations without this data. Inconsistent mass balances may directly affect the model performance, e.g. it should be ‘-I_off_ …’ in (12)-(36) and similar Equations, and ‘- h * S_out_^h^ *I_free_ * I_on_ + h * [IS] …’ in (15) and similar.

2) Mechanistic assumptions: Clarification is needed with respect to the particular choices of mechanisms and their underlying assumptions because these could bias subsequent analysis results. Examples include (a) quasi steady-state assumptions in MM and SI models, (b) assumptions on distinct gates G1/2 in compartment models, and (c) mathematical description of catalytic events in compartment models. These assumptions differentiate between models not only in the dimensions discussed in the text but, for example, also between compartment and ‘simple’ models (the latter without product inhibition); in this sense the models (and mechanistic implications) are not hierarchical as suggested.

3) Relations between mechanisms and parameters: The model-based interpretation of data (e.g., s- and i-proteasome comparison), to a large extent, relies on posterior parameter distributions, where for each proteasome substrate these distributions have been inferred independently. However, many parameters such as maximal capacity C, transport enhancement after binding (*X*_*enh*_), and diffusion in the chamber (τ) should be (nearly) identical for all substrates if the assumed physical mechanisms hold. After addressing 1) and 2) such constraints should be incorporated.

4) Model selection/validation/invalidation against experimental data: Most importantly, the (statistical) evidence for/against individual models is weak in many cases for the following reasons: (a) According to [Supplementary-material SD4-data] Equation ([37] ), (estimated) measurement noise is not included in defining the distance between model and data, making all model selection relevant only with respect to fit to the (scaled) average trajectories, where instead the probability P(model|data) under an appropriate measurement model should be computed; (b) Bayesian model selection as shown in Figure 5 concerns only the compartment models with similar numbers of parameters, but not the significantly lower-dimensional MM and SI models – a consistent selection framework is missing; (c) for key validation experiments the experimental data is either not shown (Figure 5), or predictions and experimental data are significantly different (Figure 6), both of which do not lend evidence to the overall claim of a validated quantitative model.

5) The paper needs to be carefully edited for brevity, and the Introduction should come to point much more quickly than it does at present. In particular, we ask you to remove the more philosophical discussion of the interplay between modeling and experiment in biology, that are currently part of the Introduction, including the associated conceptual schematic diagram. While we do not dispute the points you make, this paper is about a particular topic, that of modeling the reactions mediated by the proteasome, which we find quite interesting in its own right. The more global issues of modeling are not addressed by your work, and belong more appropriately in a review article.

---

## [Author Response]

The manuscript's biological conclusions predominantly rely on the interpretation of dynamic experimental data (which appear to be of good quality) with dynamic mathematical models, including Bayesian model selection for the inference of potential mechanisms. While therefore being central to the work, the modeling component raises the following concerns:

*1) Model descriptions: The presentation in*
[Supplementary-material SD4-data]*, section 2 is partially incomplete and it contains errors that may impact the overall conclusions (for detailed assessment, the authors should provide models and experimental data in appropriate numerical exchange formats). Major missing data are the initial conditions for all models/simulations, e.g. assumptions on numbers of gates (G*_*i*_*) cannot be assessed/compared between simulations without this data. Inconsistent mass balances may directly affect the model performance, e.g. it should be ‘-I*_*off*_ …*’ in (*[12]*)-(36) and similar Equations, and ‘- h * S*_*out*_^*h*^
**I*_*free*_
** I*_*on*_
*+ h * [IS]* …*’ in (*[15]*) and similar.*

We corrected and completed the model equations in [Supplementary-material SD4-data], section 2. The incorrect mass balance expressions (and several other typos) were mistakes in typing the model equations in [Supplementary-material SD4-data] and in the main text, but not in the model code itself. Because of that we only had to correct the documentation and did not have to repeat the entire model inference. The model is now provided in SBML format as Source code and will also be submitted to the Biomodels database. We have also uploaded all experimental data used for model selection and parameter inference to the Dryad Digital Repository. We will furthermore make CUDA implementations available via the ABC-SysBio website.

Furthermore, we have added a section in Supplement file 1 (“Initial conditions”) detailing the initial conditions, and we have extended the section “In silico predictions” to describe techniques used for simulations based on posterior parameter distributions.

2) Mechanistic assumptions: Clarification is needed with respect to the particular choices of mechanisms and their underlying assumptions because these could bias subsequent analysis results. Examples include (a) quasi steady-state assumptions in MM and SI models, (b) assumptions on distinct gates G1/2 in compartment models, and (c) mathematical description of catalytic events in compartment models. These assumptions differentiate between models not only in the dimensions discussed in the text but, for example, also between compartment and ‘simple’ models (the latter without product inhibition); in this sense the models (and mechanistic implications) are not hierarchical as suggested.

We have detailed the main assumptions in the results section when developing the different mechanistic models. Furthermore, we have included all the underlying assumptions in [Supplementary-material SD4-data]. More specifically:

(A) We employ the quasi-steady state assumption for the MM and SI models in order to reduce the number of model species to be numerically simulated. This approximation has been shown to be a valid representation the enzymatic models (Grima, 2009; [18]; [45]; Schnell, 2014), that does not loose information under the conditions that we have in our experimental setup. The quasi-steady state assumption holds if the catalytic rates are small compared to the on and off rates of ligand binding. For all proteases studied so far this assumption has been found to be valid; the cleaving action of the active site is complex (reviewed in (Liepe et al., 2014)), and the effective rate low; the on and off binding reactions at room temperature are fast. Of all the model reduction techniques discussed in the literature, the Michaelis-Menten mechanism has been explored and validated most completely(Gómez-Uribe et al., 2008; Schmidt et al., 2008; Smadbeck and Kaznessis, 2012). There are good pragmatic reasons for using suitably simplified models; more complicated models (e.g. those not relying on the quasi-steady-state assumption) tend to be plagued by non-identifiable parameters (Chis et al., 2014; Raue et al., 2011). We found this to be the case, especially for full model descriptions of the SI model. Such non-identifiabilities can (and do) result in numerical errors for certain parameter combinations. The latter is a known problem when exploring the model parameter space (Engl et al., 2009; Erguler et al., 2011; Rué et al., 2010).

We have also carefully corrected the equations in [Supplementary-material SD4-data] for the SI model. The SI model includes substrate and product inhibition, based on the two-site modifier scheme by [46]. The model description is now corrected.

(B) The same quasi-steady state assumption is, for the reasons outlined above, made use of in the remaining models. The models in Figure 2 had to be revised. Even though we displayed them correctly in Figure 2, we have revised the equations. Initially these two models were based on the MM model with the additional positive feedback loop. However, we agree with the referees that considered in this way these models would not be hierarchical extensions of a simpler model. We have now changed these two models so that they are based on the SI model with an additional positive feedback loop. The equations are corrected and the model selection was repeated based on the corrected model descriptions. In this way the first 4 models are hierarchical, starting from the simplest MM model, via the SI model and finally the most complex models with positive feedback loops.

We then employ again the SI model in order to describe the catalytic events in the compartmentalized models, i.e. the compartmentalization is a further level of complexity added to the simpler models. Now all discussed models are presented consistently and coherently, making their assumptions and the reasons for their development more explicit.

The description of the active site behaviour lacked detail for several models. We have therefore included much more detailed descriptions of the active site events for the non-compartmental and especially for the compartmentalized models in the main text.

(C) In the previous version of manuscript the description of the gate mechanisms lacked detail, in particular about the gate characteristics and the related transport processes. These points are now addressed fully. For example, it is now made clear that the gates are not distinct: each proteasome molecule has two gates, both of which have exactly the same characteristics. They both contain an outer area (G1) accessible for S_out_ and P_out_, and an inner area (G2) accessible for S and P. Each peptide molecule can enter and leave the proteasome chamber via the same gate, or enter via one and leave via the other gate. In the schematics of Figure 2, for clarity and convenience, we draw the peptide influx always on the left gate and the peptide efflux always on the right gate. A note to this extent is now included in the caption of Figure 2.

*3) Relations between mechanisms and parameters: The model-based interpretation of data (e.g., s- and i-proteasome comparison), to a large extent, relies on posterior parameter distributions, where for each proteasome substrate these distributions have been inferred independently. However, many parameters such as maximal capacity C, transport enhancement after binding (*X_enh_*), and diffusion in the chamber (τ) should be (nearly) identical for all substrates if the assumed physical mechanisms hold. After addressing 1) and 2) such constraints should be incorporated.*

We have inferred all the substrate-related parameters independently in order to introduce as few assumptions as possible. We agree that for a given substrate the maximal capacity could potentially be similar for different proteasome isoforms. Indeed, when inferring the parameters for s- and i-proteasomes we find that the marginal posterior distributions for C are the same. However, incorporation of several other subunits into a multimer may drastically change the kinetic properties of the complex as a whole (see for example Cardozo, J Biol Chem. 1998 Jul 3;273(27):16764-70). Furthermore, the maximal capacity is dependent on the substrate volume, and the electrostatic properties of the different substrates (which differ considerably for the three substrates). Since the short fluorogenic substrates LLVY-MCA, LLE-MCA and VGR-MCA differ in their amino acid composition and therefore in their respective approximate volumes, we expect that the numbers of substrate molecules that fit into the proteasome chamber will also differ. We have expanded Box 2 of section 4 in the [Supplementary-material SD4-data] in order to clarify this. In particular, we have added a better definition of the maximal capacity in Box 2 as well as in the main text.

Regarding the *X*_*enh*_: this describes the enhancement of the gating rate after binding of the substrate to the regulatory site. We agree that if we assume that all three tested substrates bind to exactly the same regulatory site with the same kinetics then *X*_*enh*_ should be the same for any given proteasome isoform. However, because we cannot exclude that different substrates might bind with different kinetics to potentially different proteasome isoforms we prefer to infer *X*_*enh*_ separately for the three substrates. We furthermore have to allow for the possibility that *X*_*enh*_ might differ between proteasome isoforms. By hindsight (and in agreement with the reviewers’ intuition), however, our inference results in identical marginal posterior parameter distributions for *X*_*enh*_ when comparing s- and i-proteasomes, suggesting that *X*_*enh*_ is indeed conserved between these two isoforms.

The parameter τ describes how efficient a substrate is transported inside the proteasome chamber. Therefore, τ is determined by the structure and partial charges of the inner proteasome cavity as well as the sizes and partial charges of the substrates. The inner cavity is mainly positively charged. The three investigated substrates differ in their partial charges (LLVY: 0, LLE: -1, VGR: +1). These partial charges can allow the substrates to be transported faster or slower. To test for this we inferred the parameter τ independently for each substrate and isoform. We do not detect any significant differences for τ between s- and i-proteasomes. But we find that LLVY is transported more efficiently than LLE and VGR, likely because of the absence of partial charges on LLVY, and therefore this substrate has fewer interactions with the inner proteasome cavity.

If we would have included the suggested constraints we would not have been able to detect the latter differences in substrate transport. Furthermore, inferring C and *X*_*enh*_ independently allowed us to confirm that they are indeed the same for the different proteasome isoforms.

Additionally, we have attempted to infer the parameters in a more constrained fashion for all substrates and proteasome isoforms, but we did not obtain sufficiently good model fits.

In the revised manuscript we have included a statement about the identically inferred maximal capacity for a given substrate and about the *X*_*enh*_.

*4) Model selection/validation/invalidation against experimental data: Most importantly, the (statistical) evidence for/against individual models is weak in many cases for the following reasons: (a) According to*
[Supplementary-material SD4-data]
*Equation (*[37]
*), (estimated) measurement noise is not included in defining the distance between model and data, making all model selection relevant only with respect to fit to the (scaled) average trajectories, where instead the probability P(model|data) under an appropriate measurement model should be computed; (b) Bayesian model selection as shown in*
Figure 5
*concerns only the compartment models with similar numbers of parameters, but not the significantly lower-dimensional MM and SI models – a consistent selection framework is missing; (c) for key validation experiments the experimental data is either not shown (*Figure 5*), or predictions and experimental data are significantly different (*Figure 6*), both of which do not lend evidence to the overall claim of a validated quantitative model.*

a) We have revised the model selection scheme. First of all, we have repeated the entire model selection procedure by taking into account the variability of repeated experiments. The experimental deviations increase with increasing concentrations. For this reason we have decided to use the following distance function in the ABC-SMC framework:

d=∑i(xi+ϵxi−xi*)2xi*

x* denotes the experimental data, x the simulation results and ε is a random variable drawn from a normal distribution with mean 0 and variance 0.1 (error term); this approach has previously been proposed by [53]. In this way the simulations are perturbed before being compared to the mean of the experimental data. While the details of the model fits and the intermediate model probabilities are changed using this revised distance function, the final winning model remains the same for all pairwise model comparisons.

b) We have also extended the Bayesian model selection scheme to include all models (therefore the model numbering has changed in the revised manuscript). We start with the first 4 models (lower dimensional non-compartmentalized models) and have performed model selection simultaneously. This is feasible because the number of parameters in these models is small enough in order to efficiently explore the model and parameter sample space simultaneously. The chosen model (M4) is then compared to the simplest compartmentalized model (M5). The winning model M5 is then further tested using the pairwise model comparison scheme as previously.

All relevant figures and figure supplements were revised for the new schematic and results.

c) With permission from the authors of Kisselev et al. (25) we have reproduced figures showing some of the results of their paper. These figures are now included in Figure 4. Our in silico predictions are based on fits to Suc-LLVY-MCA and Bz-VGR-MCA degradation by mouse proteasome. The experimental data by Kisselev et al. were, however, generated using rabbit muscle and yeast proteasome, respectively. Furthermore they did use the peptide Boc-LLR-MCA instead of Bz-VGR-MCA, and Suc-FLF-MNA instead of Suc-LLVY-MCA as a regulator peptide. There are a host of other potential differences related to the type of proteasome preparation, the used buffer and further experimental conditions. Because of these differences in experimental setup we did never expect an exact quantitative agreement between the results published by Kisselev et al. and our in silico predictions. We were rather hoping for qualitative agreement, confirming the basic model mechanisms, which we did indeed find.

The newly added figure should allow for easier comparison. We furthermore removed the in silico predictions based on Z-LLE-MCA, because we do not have experimental data that these predictions can be compared to, and this should further clarify our presentation. We have also included a statement in the main text that sets out the expected quantitative differences.

In Figure 6 (which is now Figure 5) we have revised the in silico predictions for the proteasomal substrate degradation in the presence of Rpt peptides. In the previous version we simulated the Rpt experiments simply by multiplying the *v*_*in*_ rate by a factor. This factor was the same for all three substrates and it was not calibrated in any way against known or measurable Rpt properties. This is a very simple approximation of the functioning of Rpt peptides, which in reality should have a binding kinetic to the alpha-rings, and upon binding, induce a conformational change that results in an increased *v*_*in*_ rate for each substrate. Furthermore, since Rpt is changing the gate conformation, the rate *v*_*out*_ might also be influenced, i.e. increased by a factor. In the revised version we calibrated these factors for each of the three substrates and now obtain quantitative similar time courses of the in silico predictions and the experimental results. Even though this way we have explored the mechanism of Rpt in more detail, there are still many other characteristics of Rpt peptides that can alter the hydrolysis, but which are ignored in the in silico predictions. For example, Rpt peptides might themselves be transported inside the proteasome chamber and be in competition with the actual substrates for chamber capacity and/or active site binding. But as much of the function of the Rpt peptides remains unknown we have taken a parsimonious description of their action and tried to minimize the

The resulting in silico predictions are now plotted including the mean and the 5%- and 95%-iles. We have also added standard deviations to the experimental data. We find that the revised in silico predictions are in excellent quantitative agreement with the experimental in vitro data (which is now clearer in the new graphical representation) and therefore validate our mechanistic model and the detected differences between s- and i-proteasomes.

5) The paper needs to be carefully edited for brevity, and the Introduction should come to point much more quickly than it does at present. In particular, we ask you to remove the more philosophical discussion of the interplay between modeling and experiment in biology, that are currently part of the Introduction, including the associated conceptual schematic diagram. While we do not dispute the points you make, this paper is about a particular topic, that of modeling the reactions mediated by the proteasome, which we find quite interesting in its own right. The more global issues of modeling are not addressed by your work, and belong more appropriately in a review article.

We have edited the paper in order to shorten it. In particular, we have removed the part that described the interplay between experiments and modeling (including the figure) from the Introduction. Furthermore we have shortened the Discussion, and we have tried to make the presentation of the results more concise without sacrificing clarity.

References:

Chis, O.-T., Banga, J.R., and Balsa-Canto, E. (2014). Sloppy models can be identifiable. q-bio.MN.

Engl, H.W., Flamm, C., Kügler, P., Lu, J., Müller, S., and Schuster, P. (2009). Inverse problems in systems biology. Inverse Problems 25, 123014.

Erguler, K., Stumpf, M.P.H., and Stumpf, M.P.H. (2011). Practical limits for reverse engineering of dynamical systems: a statistical analysis of sensitivity and parameter inferability in systems biology models. Mol Biosyst 7, 1593–1602.

Gómez-Uribe, C.A., Verghese, G.C., and Tzafriri, A.R. (2008). Enhanced identification and exploitation of time scales for model reduction in stochastic chemical kinetics. J Chem Phys 129, 244112.

Grima, R. (2009). Investigating the robustness of the classical enzyme kinetic equations in small intracellular compartments. 3, 101

Raue, A., Kreutz, C., Maiwald, T., Klingmuller, U., and Timmer, J. (2011). Addressing parameter identifiability by model-based experimentation. IET Syst Biol 5, 120–130.

Schmidt, H., Madsen, M.F., Danø, S., and Cedersund, G. (2008). Complexity reduction of biochemical rate expressions. 24, 848–854.

Schnell, S. (2014). Validity of the Michaelis-Menten equation--steady-state or reactant stationary assumption: that is the question. Febs J. 281, 464–472.

Smadbeck, P., and Kaznessis, Y. (2012). Stochastic model reduction using a modified Hill-type kinetic rate law. J Chem Phys 137, 234109.